# Identification and Functional Characterization of WRKY, PHD and MYB Three Salt Stress Responsive Gene Families in Mungbean (*Vigna radiata* L.)

**DOI:** 10.3390/genes14020463

**Published:** 2023-02-10

**Authors:** Shicong Li, Jinyang Liu, Chenchen Xue, Yun Lin, Qiang Yan, Jingbin Chen, Ranran Wu, Xin Chen, Xingxing Yuan

**Affiliations:** 1College of Life Sciences, Nanjing Agricultural University, Nanjing 210000, China; 2Institute of Industrial Crops, Jiangsu Academy of Agricultural Sciences/Jiangsu Key Laboratory for Horticultural Crop Genetic Improvement, Nanjing 210014, China

**Keywords:** mungbean, transcription factors, WRKY, PHD, MYB, synteny analysis, RNA-seq, differentially expressed genes

## Abstract

WRKY-, PHD-, and MYB-like proteins are three important types of transcription factors in mungbeans, and play an important role in development and stress resistance. The genes’ structures and characteristics were clearly reported and were shown to contain the conservative WRKYGQK heptapeptide sequence, Cys4-His-cys3 zinc binding motif, and HTH (helix) tryptophan cluster W structure, respectively. Knowledge on the response of these genes to salt stress is largely unknown. To address this issue, 83 VrWRKYs, 47 VrPHDs, and 149 VrMYBs were identified by using comparative genomics, transcriptomics, and molecular biology methods in mungbeans. An intraspecific synteny analysis revealed that the three gene families had strong co-linearity and an interspecies synteny analysis showed that mungbean and *Arabidopsis* were relatively close in genetic relationship. Moreover, 20, 10, and 20 genes showed significantly different expression levels after 15 days of salt treatment (*p* < 0.05; Log_2_ FC > 0.5), respectively. Additionally, in the qRT-PCR analysis, *VrPHD14* had varying degrees of response to NaCl and PEG treatments after 12 h. *VrWRKY49* was upregulated by ABA treatment, especially in the beginning (within 24 h). *VrMYB96* was significantly upregulated in the early stages of ABA, NaCl, and PEG stress treatments (during the first 4 h). *VrWRKY38* was significantly upregulated by ABA and NaCl treatments, but downregulated by PEG treatment. We also constructed a gene network centered on the seven DEGs under NaCl treatment; the results showed that *VrWRKY38* was in the center of the PPI network and most of the homologous *Arabidopsis* genes of the interacted genes were reported to have response to biological stress. Candidate genes identified in this study provide abundant gene resources for the study of salt tolerance in mungbeans.

## 1. Introduction

The mungbean (*Vigna radiata* L.), belonging to the Vigna Savi in the leguminosae family, contains 22 chromosomes [1]. As a functional food raw material, mungbean is rich in nutrients, including proteins, a variety of human essential amino acids, carbohydrates, dietary fiber, and bioactive substances. With the improvement in people’s living standards and the demand for healthy eating, mungbeans have a high nutritional value as an important edible bean and meet these needs of the people [2]. Salt damage seriously restricts the growth and production of crops; it is interesting that transcription factors play an important role in salt stress as intrinsic regulators of plants. Salt stress generally induces a series of genetic changes in plants, including changes in dehydration- and permeation-related aspects, causing an imbalance in Na^+^ and K^+^ concentrations and affecting plant growth and development. However, the research on mungbeans is not very in-depth, so it is very significant to analyze the salt stress-related transcription factors of mungbeans.

PHD, WRKY, and MYB are three important families of transcription factors in plants with unique functions and structures, and are indispensable in regulating signal transduction mechanisms in plants [3,4,5,6,7,8,9,10]. The PHD gene family contains a 50–80 amino acid conservative motif, mainly encoding homeodomain finger proteins, which binds to two zinc ions to maintain its stability, and its characteristic structure is Cys4-His-Cys3 [11,12]. WRKY TFs are plant-specific and one of the largest transcription factors with highly conserved WRKY domains [13,14], which are classified into three classes based on the number of conserved domains and zinc finger motifs [15,16], and regulate the expression of related genes mainly through the specific recognition of W-box (TTGACC/T) cis-regulatory elements. However, it functions through the WRKY domain [13,15]. The MYB gene family is the second largest TF superfamily in plants, its highly conserved motif is mainly located in the N-terminus, and the C-terminus belongs to the variable regulatory region [17,18,19,20,21,22]. The MYB domain generally contains 50 amino acid residues [23] and has three repeat types, R1, R2, and R3 [21,24], which can be divided into four subclasses according to the number of repeat types [21]. These classification standards provide a good reference for similar research on mungbeans.

Previous research reports have shown that PHD transcription factors have abundant functions. For example, the barley *HvMS1* gene can regulate temperature-sensitive sterility as a PHD finger-like transcription factor [25] from bamboo PHD genome-wide screening of genes for 16 abiotic stress response genes [26]. Rice PHD-like protein OsTTA is a constitutive expression regulator of multiple metal transporter genes and regulates plant growth and development [27]. The zinc finger protein *Arabidopsis VIL1* regulates drought response through the ABA pathway [28]. WRKY is a widely studied transcription factor in plants. In maize, *ZmWRKY20* and *ZmWRKY115* transcription factors have been found to interact with each other in the nucleus to regulate salt tolerance [8]. It was previously reported that the *GmWRKY21* transcription factor in soybean regulates aluminum stress tolerance [29]. *WRKY63* participates in vernalization induction by binding to the promoters of flowering sites in *Arabidopsis* [30]; furthermore, tomato *SlWRKY37* delays leaf yellowing by improving plant sensitivity to external senescence signals [31]. MYB TFs have been extensively studied in plants; for example, *MdMYB2* can activate an E3 ubiquitin ligase *MdSIZ1* in apples and promote the synthesis of anthocyanins, thus improving the cold tolerance of apples [32]. Three *OsMyB36* TFs in rice are involved in the regulation of *OsCASP1* expression, which affects CS synthesis and the selective absorption of mineral elements in roots [33]. The degradation of *OsMYBc* by ubiquitinated protein *OsMSRFP* prevents *OsHKT1* from interacting with *OsMYBc* to regulate Na ^+^ transport during salt stress in plants [34]. However, salt stress tolerance genes have rarely been reported in mungbeans by comparative genomics and transcriptome analyses. Mining candidate genes would be significant for the genetic improvement of mungbeans.

The research into transcription factors has led to great progress in plant salt stress regulation pathway research; it has been reported that DRE/CRT cis-acting elements can promote the ABA-independent gene expression so as to respond to dehydration stress. Dehydration and high salt stress treatments in *Arabidopsis thaliana* lead to the binding of *DREB2 A* and *DREBB* to DRE/CRT, so that these genes can be strongly induced in roots [35]. In *Arabidopsis*, salt-induced synthesis of carotenoids was also found to provide sufficient ABA precursors to ensure continuous ABA synthesis and enhance plant salt tolerance [36]. Although abscisic acid (ABA) is involved in mediating osmotic stress in plants, other studies have found that an ABA-independent salt stress response pathway exists in *Arabidopsis* ABA mutants, which could strongly induce the expression of *NCED3*, *AAO3*, and *ABA1* genes under salt stress [37]. In the literature, *AtNCED3* is induced in the vascular wall of *Arabidopsis* under drought stress [38], which may be due to the important regulatory role of ABA-dependent osmotic stress in both salt stress and drought stress. Previous studies have shown that plant vacuolar NHX protein plays a transport function in the process of Na^+^/H^+^ and K^+^/H^+^ exchange [39,40], mainly transporting intracellular Na^+^ to vacuoles and maintaining a low intracellular Na^+^/K^+^ ratio [41], thus regulating plant salt tolerance and K^+^ nutrition. In contrast to vacuolar NHX proteins, there are also Na^+^/H^+^ antiporters *NHX5* and *NHX6*, which may be involved in the maintenance of organelle pH and ion homeostasis [42] and may play an important role in the regulation of salt stress. The ABA synthesis pathway has gradually become a hot research topic in plant salt tolerance; a recent study has found that OsCSLD4 is a cellulose synthase protein in the rice cell wall which can enhance the expression of ABA synthesis genes and improve the salt tolerance of rice [43]. Transcription factors contain corresponding cis-acting elements that can mediate the expression of ABA-responsive genes, including WRKY, MYB, AP2/ERF, and HDzip TFs, which are induced by salt stress in many plants [44]. For instance, the ERF1 transcription factor could bind to the promoter of *OsABI5*, a downstream gene of the ABA signaling pathway, and inhibit its expression, thereby enhancing rice seed germination under salt stress [45]. These studies indicate that the ABA-dependent salt stress response pathway can regulate salt stress in a variety of plants, in which a series of abiotic stress response genes, including transcription factors, play an important role; however, few genes that respond to ABA have been studied in mungbeans.

In general, the early stage of salt stress mainly represses the growth of the plant root; a previous study showed that the roots of the tested plants were significantly inhibited by high salt treatment for about four days. The endodermal signal network of the roots of salt-stressed vascular plants regulates the growth of seedling roots, mainly through the ABA-dependent pathway to inhibit the development of lateral roots and reduce the salt contact area of roots [46]. The root structures of 31 different *Arabidopsis* plants are affected by salt stress. A dynamic analysis of root structures under salt stress showed that the growth of root branches and primary and lateral root meristems had different regulation strategies to salt stress [47]. We hypothesized that changes in the root structure under salt stress play an important role in plant salt tolerance. A genome-wide association analysis of *Arabidopsis* 330 samples under both salt stress and phosphorus starvation identified 12 loci in which phosphorus starvation interferes with the lateral root salt stress response [48]. Next, a genome-wide association analysis of gene loci related to root structure under salt stress was carried out to illustrate the effect of salt stress on the root structure. In a salt-related GWAS analysis, the relevant results showed that 347 *Arabidopsis* samples, associated with 100 loci of the root system structure, including *CYP79 B2* and *CYP79 B3*, under salt stress are positively correlated with lateral root development [49]. Root transcriptome data provide good material for studying related gene families in mungbeans. According to the above studies, it can be also concluded that the study of the genetic mechanism of root salt stress proves that the effect of salt stress on root development is mainly manifested in lateral roots.

To address the above issues, in this study, transcription factors of WRKY, PHD, and MYB in mungbeans were identified and 83, 47, and 159 genes were obtained, respectively. Sequence alignment and a phylogenetic tree analysis of these genes enabled us to further understand the functions of the three gene families of mungbeans. Seven differentially expressed genes in mungbean roots were screened by transcriptome sequencing. In addition, real-time fluorescence quantitative PCR was used to verify that these seven candidate genes not only responded to salt stress, but also responded to ABA and PEG treatment to varying degrees, which indicated that the seven candidate genes may have different regulatory pathways in response to salt stress. The interaction proteins of five candidate TFs were predicted by protein–protein interaction analysis (PPI), and a homologous alignment analysis of mungbean interaction genes found that *Arabidopsis* homologous genes had important regulatory roles in abiotic stress, which provided clues for us to study the salt stress response pathway of mungbean candidate genes and laid a theoretical foundation for the cultivation of salt-tolerant varieties of mungbeans. Additionally, the relevant results are useful for mungbean salt tolerance improvement and gene function identification.

## 2. Materials and Methods

### 2.1. Identification of Three Gene Family Members in Mungbean

The WRKY, PHD, and MYB protein sequences of mungbeans were obtained from the genome database of Legumes Laboratory of Jiangsu Academy of Agricultural Sciences [50]. All the data (Illumina, Nanopore, Hi-C and RNAseq) that support the findings of this study are openly available in the NCBI SRA (Sequence Read Archive) database under the Bioproject ID: PRJNA660308 [50]. Firstly, three gene family members containing WRKY (PF03106), Znf_PHD (IPR001965), and HTH_MYB domains were identified using pfam (PF03106), pfam (IPR001965), and pfam (PS51294) protein motifs [51], respectively. Then, the HMMER 3.0 (https://www.ebi.ac.uk/Tools/, accessed on 25 January 2022) tool was used for further analysis of the three screened gene family members using the hidden Markov model (HMM); the threshold of the valued E-group was 1 e-05. Finally, the SMART (http://smart.embl-heidelberg.de/, accessed on 25 January 2022) online tool was used to verify the conserved domains of the selected gene family members [52].

### 2.2. Evolutionary Tree, Conserved Motifs, Gene Structure, and Promoter Analysis

The ClustalW program and the MEGA7 [53,54] software was used for multiple sequence alignment and evolutionary tree analysis. The calculation method was NJ and the parameter was set to 1000 repeated iterations. A conservative motif analysis of the three gene families was performed using MEME (5.5.1) online software (https://meme-suite.org/meme/tools/meme, accessed on 25 January 2022) [55]. The conserved motifs and gene structures were visualized by using the Gene Structure View plugin in TBtools software. The genome annotation files were in in gff3 format for the mungbean and the results of the MEME conserved motif analysis [56]. The 2000 bp upstream of ATG was extracted from the genome file in fasta format and the genome annotation file in ggf3 format as the promoter sequence, and the corresponding sequences of WRKY, PHD, and MYB gene families were input into the PlantCARE database website (http://bioinformatics.psb.ugent.be/webtools/plantcare/html/, accessed on 25 January 2022) for cis-acting element analysis [57]. The results were visualized by TBtools software.

### 2.3. Chromosomal Position, Gene Duplication, and Collinearity Analysis of Genes in the Same Gene Family

The genome database of mungbeans sequenced and assembled by our laboratory was used to locate the VrWRKYs, VrPHDs, and VrMYBs on chromosomes. The positions of each family gene on 11 chromosomes of mungbeans were obtained. The visual analysis tool for chromosome localization was TBtool software.

The GFF3/GTF Gene Position (Info.) Parse program was used to obtain the position of the gene on the chromosome in the mungbean genome annotation file, and the collinearity analysis was performed by the One Step MCScanx program. The above steps can be run by TBtools (https://github.com/CJ Chen/Tbtools, accessed on 25 January 2022) drawing software, and the visual analysis of the graph was carried out by the Advanced Circos program of the software [56,58]. The intraspecific collinearity of mungbeans and the interspecific collinearity between mungbeans and *Arabidopsis thaliana* were analyzed. Ka/Ks values were calculated using TBtools [56].

### 2.4. Plant Material Treatment and Gene EXPRESSION Pattern Analysis

Four mungbean seedlings, which were salt-tolerant varieties A1 and A3 (accessions no. G109 and G146) and salt-intolerant varieties C1 and C2 (accessions no. G339 and G340), grown to 8 days were treated with NaCl at concentrations of 0 mM, 100 mM, and 200 mM. After about 10 (T1) and 15 (T2) days of salt treatment, samples were taken from each treatment group and total RNA was extracted, and then transcriptome sequencing was performed [59,60]. The expressions of differentially expressed genes in VrWRKYs, VrPHDs, and VrMYBs were analyzed. The genes with a *p*-value < 0.05 were screened out, and the change fold of log2 was used to express the response of differentially expressed genes to salt stress, which was presented in the form of a heat map [60].

In order to verify the differential gene expression patterns of transcriptome sequencing data, mungbean (A3, salt-tolerant) samples stored in the Cash Crop Research Institute of Jiangsu Academy of Agricultural Sciences were planted in a material room with a constant temperature of 28 °C. The seedlings were treated with ABA(100 μM), NaCl (100 mM), and 20% PEG after one week of light (16 h) and dark (8 h) culture. The roots of the seedlings under different treatments were sampled at 0 h, 4 h, 12 h, 24 h, and 48 h, and then flash-frozen in liquid nitrogen and placed in an ultra-low temperature refrigerator at −80 °C for subsequent verification of gene expression patterns. 

### 2.5. qRT-PCR Analysis of Candidate Genes of Three Gene Families under Salt Stress

A FlaPure Plant Total RNA Extraction Kit (Jingsha Biology, Beijing, China) was used to extract total RNA from mungbean roots. Reverse transcription synthesis of cDNA was performed using a UnionScript First-strand cDNA Synthesis Mix for qPCR (with dsDNase) (Jinsha Biological, Beijing, China). qRT-PCR was performed with an ABI prism 7500 real-time PCR System (Thermo Fisher, 5823 Newton Dr, Carlsbad, CA 92008, USA) and the reagent used was ChamQ SYBR qPCR Master Mix (Vazyme, Nanjing, China). The procedures used for PCR amplification are referred to in [61]. The relative expressions of candidate genes from five gene families in mungbeans were analyzed by the 2^−ΔΔCT^ calculation method. The gene-specific primers were designed by Primer 5 software and the reference gene was *VrACTIN3* (*EVM0012789*) [62]. Three biological replicates and three technical replicates were set for each group of experiments. Primers are listed in Appendix A.

### 2.6. Protein Interaction Network Analysis of Differentially Expressed Genes

The differentially expressed gene protein sequences were input to the online STRING (https://cn.string-db.org/) website. The parameter settings were as follows: minimum required interaction score: 0.4 and maximum number of interactors to show: 50. The results of predicted differentially interacting proteins were output in TSV format [63]. Cytoscape 3.9.1 software was used to further analyze the TSV file results, and the Btweenness parameter was used to show the size of nodes, thus indicating the strength of protein interactions [64]. 

## 3. Result

### 3.1. Mungbean WRKY, PHD, and MYB Three Gene Family Members

A total of 84 *VrWRKYs*, 64 *VrPHDs*, and 153 *VrMYBs* (e-value < 1 × 10^−5^) were screened from the whole genome of mungbeans by the pfam (https://pfam.xfam.org/, accessed on 25 January 2022) website. HMMER (https://www.ebi.ac.uk/Tools/hmmer/, accessed on 25 January 2022) was used to verify the conserved domain, and 83 *VrWRKYs* (Appendix A), 47 *VrPHDs* (Appendix A), and 149 *VrMYBs* (Appendix A) were further identified. As shown in Figure 1A, the WRKY gene family sequence is strongly conserved and has typical structural characteristics with the N-terminal WRKYGQK heptapeptide sequence. The PHD gene family has a zinc-binding motif similar to C4 HC3 (Cys4-His-Cys3), which is composed of multiple cysteines and one histidine. The MYB gene family contains the tryptophan cluster W primary structure and there is a 19 amino acid interval between the two tryptophans. These conserved domains represent the characteristics of the three gene families and the criteria for screening mungbean WRKY, PHD, and MYB transcription factors.

### 3.2. Phylogenetic Tree and Sequence Structure Analysis of Three Gene Families in Mungbean

In order to study the evolutionary relationship and the branching of the three gene families, we performed an evolutionary tree analysis for each gene family. In Figure 1B, VrWRKY was divided into three sub-groups, with 35, 24, and 23 genes in sub-groups I, II, and III, respectively. The cluster analysis of VrPHD showed two sub-groups, I and II, with 10 and 37 genes, respectively. VrMYB was also divided into two sub-groups, and each had 112 and 37 genes, respectively. The genes in the same branch have high homology and conservation in evolution. Figure 2A reveals that the protein domains of the WRKY gene family in the same branch are relatively similar and the protein domains in different branches are different. Among them, sub-group I contains three kinds of conserved motifs, 1, 2, and 4, which are mostly located in the C-terminus of the gene. Sub-group II contains only conserved motifs 1 and 2, and sub-group III contains conserved motif 1. It was concluded that WRKY conserved domains were generally located in the middle or C-terminus of the gene, but less in the N-terminus. Sub-group I of the PHD gene family contains five protein domains, which are evenly distributed in the N- and C-terminus of the gene. Sub-group II contains two or one of the protein domains 1 and 5, but the positions of these two conserved motifs are not the same on the gene, indicating that the function of the PHD family may be more diverse. The MYB gene family has four similar protein domains, but most of the protein domains of sub-group I are located in the N-terminus of the gene. The protein sequences of sub-group II are relatively long, with some protein domains located in the N-terminus and some in the middle of the sequence. Based on the above analysis, each gene family contains the conserved motifs in Figure 1A, but the conserved motifs in different sub-groups of the same gene family have some differences. The evolutionary tree in Figure 1 shows that mungbeans have acquired more VrWRKYs, VrPHDs, and VrMYBs during the evolution process, which has increased the diversity of the gene functions of the three gene families.

In the gene structure and cis-acting elements analysis (Figure 2A and Appendix A), it was shown that *VrPHDs* have more exons with short sequences compared to *VrWRKYs* and *VrMYBs* gene family exons. The number of introns and exons is significantly different among different branches of different gene families, and there are few differences among the same branches. This phenomenon leads to the multiplicity of alternative splicing in the three gene families, which leads to the diversity of gene functions. From the cis-acting elements, it can be seen that the most regulatory elements in the three gene families of mungbean are abscisic acid-responsive elements. As shown in Appendix A, VrWRKYs, VrPHDs, and VrMYBs have 72, 32, and 128 ABRE elements, respectively. Methyl jasmonate, defense stress, salicylic acid, drought, and gibberellin elements were commonly shared. It is speculated that these genes may respond to abiotic stress.

### 3.3. Collinearity Analysis and Chromosome Mapping of the Three Gene Families in Mungbeans

In the collinearity analysis, there were 44, 13, and 58 collinear gene pairs in WRKY, PHD, and MYB gene families (Appendix A), respectively, indicating that these gene pairs had higher homology. Further analysis found that there were more than two homologous gene pairs in the related genes of the three gene families, especially *EVM0004727.1* of WRKY TFs, *EVM0010532.1* of PHD TFs, and *EVM0020569.1* of MYB TFs, which implied that these genes played an important role in the evolution of mungbeans. Next, the Ka/Ks ratios of these collinear gene pairs were analyzed (the calculation results of 12 collinear gene pairs in the VrPHDs gene family can be found in Appendix A), and we found that all the Ka/Ks values were far less than 1, which indicated that the above *VrPHDs* experienced strong purification selection. As in Table 1 and Appendix A, 40 VrWRKYs gene pairs experienced fragment duplication and purification selection. As shown in Figure 3, we conducted an interspecies collinearity analysis of mungbeans and *Arabidopsis*; there are 57, 27, and 81 collinearity gene pairs in VrWRKYs, VrPHDs, and VrMYBs between the two species (Table 1 and Appendix A), respectively. The number of these gene pairs is more than half the number of genes in three corresponding gene families of mungbeans, which indicates that mungbeans and the model plant *Arabidopsis* have a relatively close homology.

Chromosome locations were identified for the three gene families of mungbeans by means of the mungbean genome annotation file gff3 and Gene Location Visualize GTF/GFF tools [56]. The results in Figure 4 show the chromosome distribution of the three gene families. Each gene family was distributed on 11 mungbean chromosomes, but the distribution was not particularly similar; only one WRKY gene was distributed on chromosome 8 and only one PHD gene was distributed on chromosomes 2 and 7. In addition, the number of genes in the same gene family varied greatly on each chromosome, and there was no inevitable connection with the chromosome length. 

### 3.4. Gene Expression and Protein Interaction Analysis of VrWRKY, VrPHD, and VrMYB Differentially Expressed Genes 

As root tissue is a good material for studying salt resistance-related genes in mungbean, The RNA-seq data of WRKY, PHD, and MYB gene families were analyzed, and seven differentially expressed genes were obtained (Figure 5A), including *EVM0009300*, *EVM0015430*, *EVM0027733*, *EVM0009300*, and *EVM0015733*. Among them, *EVM0027995* was significantly downregulated in four mungbean samples under different salt concentrations, and *EVM0009300* was the most obvious one. The expression of EVM0000108, *EVM0010532*, and *EVM0019669* decreased less than the above four genes under salt stress. More interestingly, *EVM0000108* was differentially expressed in different samples and treatments. The expression of *EVM0000108* was significantly increased in A1 material under salt stress, but significantly decreased in A3 material. The expression was only weakly downregulated in two salt intolerant materials: C1 and C2. The results indicated that *EVM0000108* might participate in the response mechanism of salt stress in different varieties.

A qRT-PCR analysis of seven candidate genes in response to salt stress was performed. As shown in Figure 6, the expression of PHD candidate genes *EVM0010532* and *EVM0027733* decreased (*p*-value = 0.0011) under salt and drought stress, indicating that the decrease in the two genes’ expression may be caused by osmotic stress caused by salt treatment. Upon treatment with exogenous ABA, the expression of *EVM0010532* and *EVM0027733* was different. The expression of *EVM0010532* was inhibited within 3–12 h, which was 0.3-fold lower than the value at 0 h (*p*-value = 0.0003), and the expression of *EVM0027733* increased by more than 0.3-fold within 12–24 h (*p*-value = 0.0007), which indicated that *EVM0010532* and *EVM0027733* may regulate osmotic stress caused by salt treatment through an ABA-dependent pathway. In *Arabidopsis*, it has been reported that the *EVM0010532* homolog *AL5* can enhance plant salt tolerance [65], but here the expression of EVM0010532 was downregulated by 0.5-fold (*p*-value = 0.002), indicating that *EVM0010532* may has a completely different regulatory pathway in response to salt stress compared with the *Arabidopsis* homolog genes. The expressions of the three *WRKY* genes were downregulated by 0.5-, 0.4-, 0.6-fold, respectively, under salt stress, especially *EVM0015430* and *EVM0027995* (Figure 5). As the homologous gene of soybean *GmWRKY54*, *EVM0019669* may have similar salt and drought tolerance functions in mungbeans [66]. *EVM0000108*, a MYB candidate gene, responded in the early stages of ABA, NaCl, and PEG stress treatment, while *EVM0009300* only responded to NaCl and ABA and its expression was downregulated by salt stress and upregulated by ABA stress. The above analysis verified that the responses of the candidate genes of the three gene families to salt stress were consistent with the transcriptome analysis data, which further indicates that salt stress affected the expression of these genes.

In order to study the possible synergistic effect of these three gene families under salt tolerance, protein interaction analyses of seven DEGs were undertaken. The analyses showed that EVM0027995 was in the center of the PPI network, and we deduced that EVM0009300, EVM0000108, EVM0019669, EVM009300, EVM0000108, EVM0019669, and EVM0015430 were highly associated with multiple genes (*p* > 0.4). Most of the genes predicted to interact with EVM0027995 are related to stress defense response, for example, EVM0027995.1 (VrWRKY38) and EVM0002826.1 (0.44), EVM0027995.1 (VrWRKY38) and EVM0002218.1 (0.44), and EVM0027995.1 (VrWRKY38) and EVM0029605.1 (0.56). For EVM0002826 (0.44), its homologous gene in *Arabidopsis* could regulate the biosynthesis of anthocyanin and play a role in stress resistance [67]. 

More importantly, as shown in Figure 5A, the gene expression of EVM0002826 in A3 decreased. Therefore, it was speculated that the response of EVM0027995 to NaCl, ABA, and PEG might be related to the interaction of EVM0002826. Moreover, its homologous gene, EVM0007730, a protein predicted to interact with EVM0000108, has also been associated with stress defense response in *Arabidopsis* [68]. EVM0007730 also responded to salt stress in A3 (*p*-value = 0.0025), indicating that the above two genes may have common regulatory pathways in stress tolerance. Furthermore, the predicted protein EVM0004554 interacting with EVM0015430 may relate to salt stress, as the homologous gene of EVM0004554 was reported to have a stress-resistance function in *Arabidopsis* [69]. The above analysis indicates the potential relationship of these genes under abiotic stress.

## 4. Discussion

### 4.1. Identification of New Members of VrWRKYs, VrPHDs, and VrMYBs in Mungbean

WRKY, PHD, and MYB transcription factors are widely found in plants [3,11,13,15,18,21], and have been thoroughly studied in a variety of plants. However, there are few studies on VrWRKYs, VrPHDs, and VrMYBs in mungbeans. This study is the first to screen the above three gene families in mungbean genomes, which plays an important role in studying the function of *VrWRKYs*, *VrPHDs*, and *VrMYBs* in mungbeans. Through the identification of gene family members, 83 VrWRKY TFs, 47 VrPHD TFs, and 149 VrMYB TFs were screened. The genes in Figure 1A all contain highly conserved DNA-binding domains, indicating that they mainly function in transcriptional regulation of genes. In particular, the transcriptional regulation of WRKY TFs on abiotic stress has been thoroughly studied in a variety of plants [9,29,70], which is helpful in our study of the function of salt stress-related genes in mungbeans.

Although WRKY, PHD, and MYB genes all contain corresponding conserved domains, they have formed different subclasses and branches during the long-term evolution of the plant, providing genes with functional diversity for plants to adapt to different environments [32,71,72]. We found that conserved domains of different branches have undergone corresponding changes; some have added new conserved motifs and some have deleted part of conserved motifs (Figure 2). It can be assumed that mungbeans have produced a large number of genes with similar structures but different functions through gene recombination or chromosome rearrangement in their long-term evolution [73]. The gene structure in Figure 2 shows that the distributions of introns and exons of the three families have typical characteristics. For instance, the numbers of exons and introns of all genes in PHD sub-group I are relatively consistent, while the numbers of exons and introns of genes in sub-group II are quite different, which leads to the diversification of genes with alternative splicing [73] and further enriches its transcriptional regulation functions [74], resulting in a strong conservation of the PHD sub-group I genes and diverse structures of the sub-group II genes. Transcription factors play an important role in plant environmental stress and are generally related to their upstream cis-acting elements [75]. As shown in Figure 2, a promoter sequence analysis showed that the upstream of *VrWRKY*, *VrPHD*, and *VrMYB* genes contains abscisic acid, methyl jasmonate, defense stress, salicylic acid, drought stress, gibgibellin, and other response elements. These cis-acting elements can initiate the expression of downstream genes when plants are subjected to external environmental stimuli [76,77]. Although the functions of the related genes can be predicted from the sequence analysis, further experiments are needed to clarify the specific regulatory mechanisms of VrWRKYs, VrPHDs, and VrMYBs in response to salt stress.

### 4.2. Evolutionary Relationships of WRKY, PHD, MYB Genes in Mungbean

In the evolutionary study of VrWRKY, VrPHD, and VrMYB gene families, as shown in Figure 1, we found that a large number of genes in each gene family have in-species collinearity. Mungbeans are diploid plants, which indicates that mungbeans have a gene copy event through chromosome replication during natural evolution [59]. The ratio of synonymous substitutions (Ka) and synonymous substitutions (Ks) in the three gene families is listed in Appendix A, which prove that these gene copy events belong to fragment replication, enrich the genetic information of three gene families in mungbeans, and improve their adaptability to external environmental stimuli. Figure 3 shows that there are a large number of gene duplication events in VrWRKYs, VrPHDs, and VrMYBs in mungbeans, which enriches the numbers of the three gene families [22,57]. From the analysis in Appendix A, it can also be concluded that there were collinearity genes in the three gene families of mungbeans, which were mostly close to each other in chromosome distance and generally occurred at the ends of adjacent chromosomes or inside chromosomes, further indicating that this phenomenon may be the result of the common influence of fragment events and conserved evolutionary strategies in mungbeans [78].

Research into the collinearity relationship between different species can help us to study the gene function of corresponding species through homologous genes and provide data support. For example, in Figure 3, by analyzing the collinearity relationship between *VrWRKY*, *VrPHD*, and *VrMYB* genes and *Arabidopsis* genes, it indicated that mungbeans and *Arabidopsis* have multiple homologous gene pairs. Despite that those genes are less known in mungbeans, its homologous genes have been widely studied in *Arabidopsis* [7,21,28], which can help us to predict the regulatory effects of the related genes in mungbeans. Of course, in order to understand which species has the closest kinship to mungbeans, we need to do more collinearity analysis between different species and mungbeans [57]. For example, soybeans, as an important edible and oilseed crop, have been widely studied and the corresponding research is in-depth [79,80,81]. We can also use a collinearity analysis between soybeans and mungbeans to provide abundant information for gene function studies in mungbeans.

### 4.3. Differential Expression Genes and Protein Interactions of WRKY, PHD, and MYB in Mungbeans under Salt Stress

Through a differential gene expression analysis (Figure 5A), a total of seven differentially expressed genes were screened (Figure 6). Additionally, these genes not only responded to salt stress, but also responded to ABA and PEG treatment to different degrees, indicating that the seven transcription factors may have different regulatory pathways in response to salt stress. A part of genes may regulate salt stress response through an ABA-dependent form [36,37,45], such as *EVM0009300*, and others may regulate salt stress response through osmotic stress. For instance, *EVM0019669* showed great response to NaCl and PEG stress, but no obvious response to ABA. To verify the results of potential regulatory pathways of candidate genes, we should further perform overexpression analyses of candidate genes in corresponding transgenic materials and analyze gene expression patterns of ABA regulation pathways and osmotic stress-related genes under salt stress [45]. We performed a protein–protein interaction (PPI) network analysis of the seven candidate genes so as to investigate their possible interaction genes (Figure 5B) [63], and attained the interaction results of five genes. Among them, the EVM0027995 protein node is the largest and has a high interaction strength with the EVM0002826 protein. In addition, *EVM0002826* is homologous to the *Arabidopsis* gene *WRK41*, which may be a regulator of a salicylic acid pathway [82], which indicates that *EVM0002826* possible plays an important role in abiotic stress. Furthermore, we supposed that the interaction between *EVM0027995* and *EVM0002826* may regulate the response of salt stress. In order to explain the interaction of the above two proteins, we need to conduct bimolecular fluorescence complementary experiments, yeast two hybrid, and so on. In addition to EVM0010532 and EVM0027733, the interaction genes predicted by the other six DEGs are functionally related to abiotic stress (Appendix A). In Figure 5A, a differential gene expression analysis of the candidate genes and the relative interaction of genes showed that these genes can all respond to salt stress, which further proves the interaction of the genes in PPI. Gene expression analyses and protein interaction experiments of these interacting genes will be more conducive to explaining the regulatory pathway of related candidate genes in the stress response pathway, which is of great significance for the salt resistant molecular breeding of mungbeans.

In this research, sequence alignments of VrWRKYs, VrPHDs, and VrMYBs were analyzed to shed light on the relationship between the sequence structure and evolution. On the basis of mungbean salt stress, we conducted transcriptome sequencing to identify differentially expressed genes in three gene families, obtaining seven stress response genes all containing ABA, GA3, salicylic acid, and other stress response elements. These results strongly support the role of these genes in salt stress regulation [76]. However, the number of differentially expressed genes is low and the change in the differentially expressed gene ratio is not high; therefore, it can be presumed that although there are differences in salt tolerance among the four mungbean samples, they are relatively salt tolerant compared to other species on the whole, which leads to the low expression of salt stress response genes. The above results provide key data for studying the molecular mechanisms of salt tolerance in mungbeans and more information for the breeding of salt-tolerant varieties of mungbeans.

## 5. Conclusions

In conclusion, 83 VrWRKYs, 47 VrPHDs, and 149 VrMYBs were screened by combining the genome data of mungbeans. The characteristics of the three gene families were analyzed in detail by conserved motifs, phylogeny, structural domains, cis-acting elements, chromosomal locations, and collinearity analyses. Combined with transcriptome sequencing under different concentrations of salt stress, seven candidate genes were screened for significant response to salt stress, including three VrWRKYs, two VrPHDs, and two VrMYBs. Further gene expression analyses by qRT-PCR showed that all seven candidate genes could respond to salt stress and also had different degrees of response to ABA and PEG, indicating that the seven candidate genes may have complex regulatory mechanisms in response to salt stress. The possible interacting proteins of five candidate genes may provide important clues for studying the salt stress gene network. This investigation into the VrWRKY, VrPHD, and VrMYB TF families’ roles in salt stress will provide excellent gene resources for salt-tolerant mungbean breeding and realize the cooperation of different gene families in salt tolerance. 

## Figures and Tables

**Figure 1 genes-14-00463-f001:**
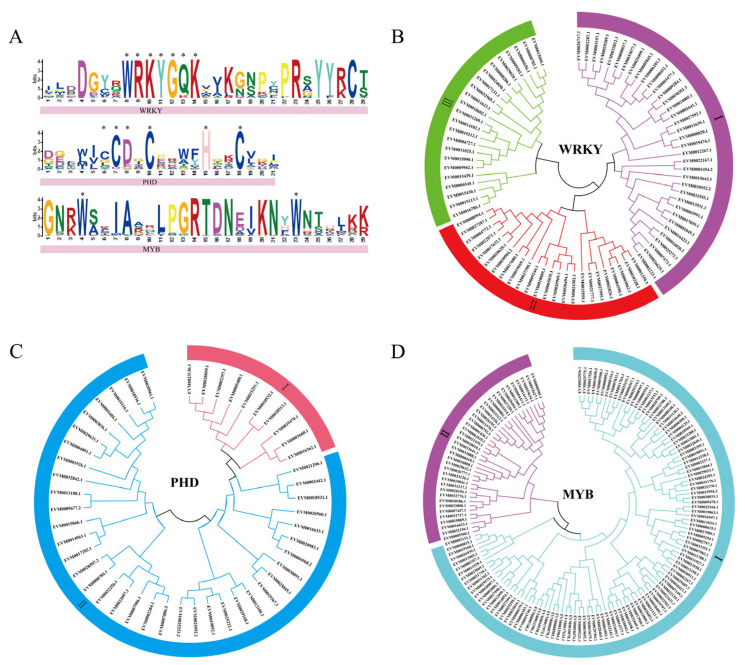
DNA conserved domains and evolutionary tree analysis of mungbean *WRKYs*, *PHDs*, and *MYBs*. (**A**) The motifs of the three gene families, in which the letter sizes of different loci in each family indicate the sequence conservation of the conserved motif (measured in bits). The size of a single letter in the letter pile indicates the size of the frequency distribution of the corresponding amino acid of the conserved motif and * in the figure indicates that the corresponding amino acid has the highest frequency distribution at this site. Phylogenetic tree of *WRKYs*, *PHDs*, and *MYBs* (**B**–**D**): each family has a different branch and each branch is shown in a different color and is marked with roman numerals such as I, II, and III.

**Figure 2 genes-14-00463-f002:**
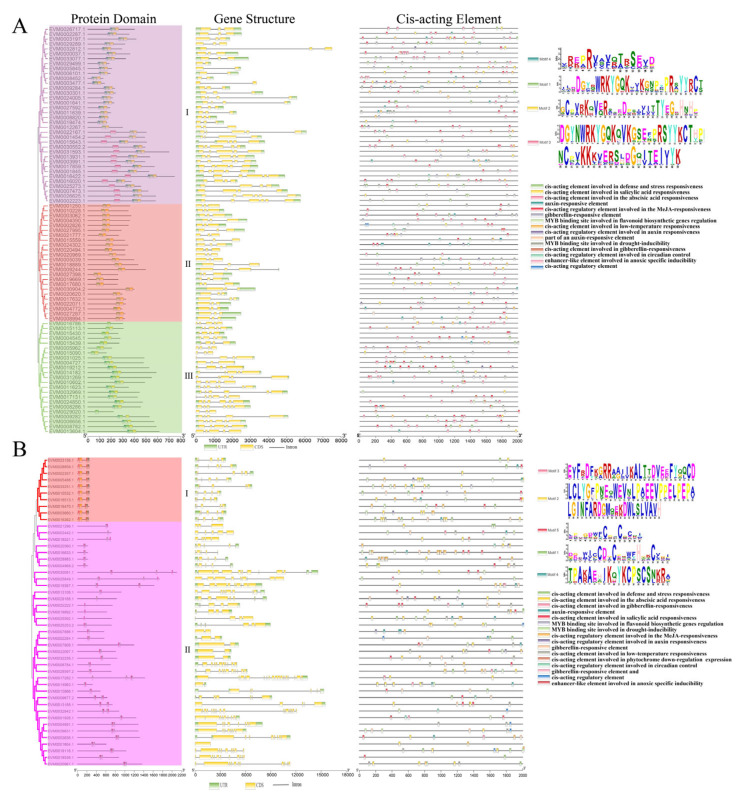
Analysis of protein domains, gene structures, and cis-acting elements. WRKY and PHD gene families motif and protein conservative domain (**A**,**B**). Cis-acting elements are located within 0 to 2000 bp upstream of the gene initiation codon.

**Figure 3 genes-14-00463-f003:**
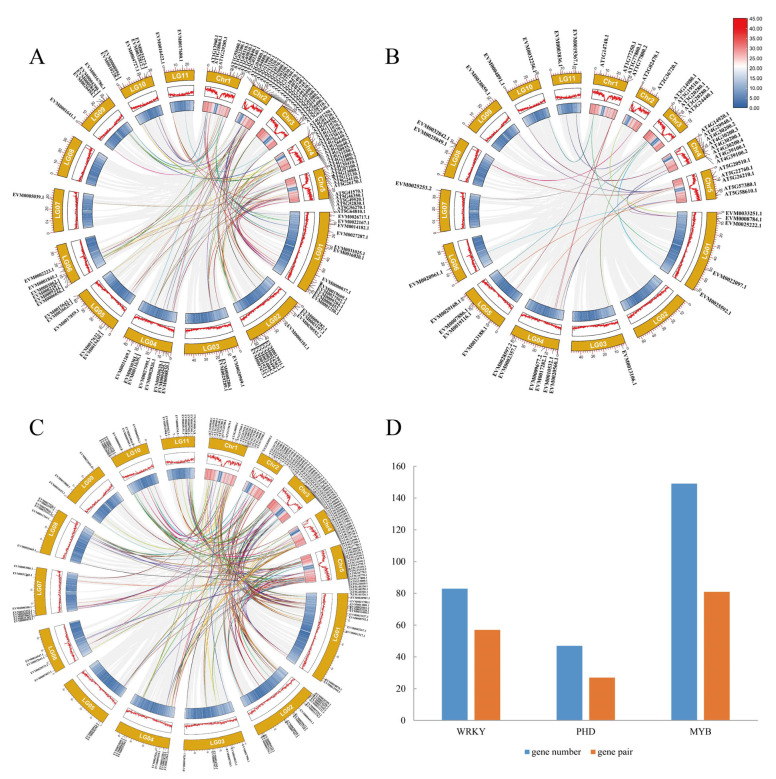
The synteny analysis between mungbeans and *Arabidopsis thaliana* of three gene families. The synteny analysis of VrWRKYs, VrPHDs, and VrMYBs, respectively (**A**–**C**). Chr1–Chr5 are the five chromosomes of *Arabidopsis* and LG01–LG11 represent the 11 chromosomes of mungbean. (**D**) The number of members and collinear gene pairs of the three gene families.

**Figure 4 genes-14-00463-f004:**
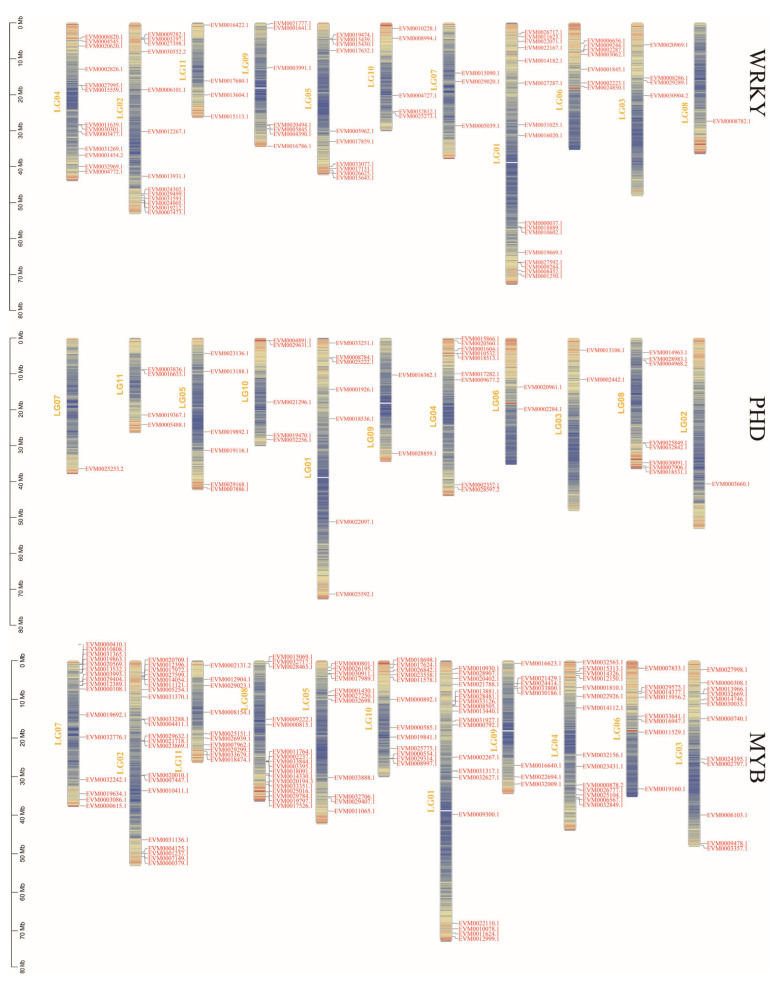
Distribution of three mungbean gene families on 11 chromosomes.

**Figure 5 genes-14-00463-f005:**
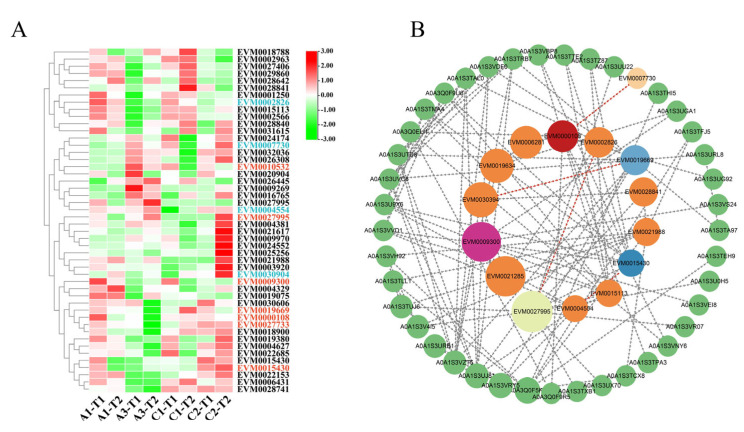
Heat map and protein interaction analysis of differentially expressed genes. (**A**) Expression profiles of WRKY, PHD, and MYB gene families. A1 and A3 represent the two salt−tolerant samples of mungbeans, C1 and C2 represent the two salt−intolerant samples of mungbean, and T1 and T2 represent the concentrations of 100 mM NaCl and 200 mM NaCl, respectively. The red gene numbers represent seven differentially expressed genes and the blue and black gene numbers represent the predicted interacting genes. Genes with FPKM > 0.5 were chosen in this study. (**B**) Circles represent the nodes of each gen, and the sizes of the circles represent the intensity of protein interactions, which is deduced by the value of the Btweenness parameter. The two lines between nodes indicate that two proteins may interact.

**Figure 6 genes-14-00463-f006:**
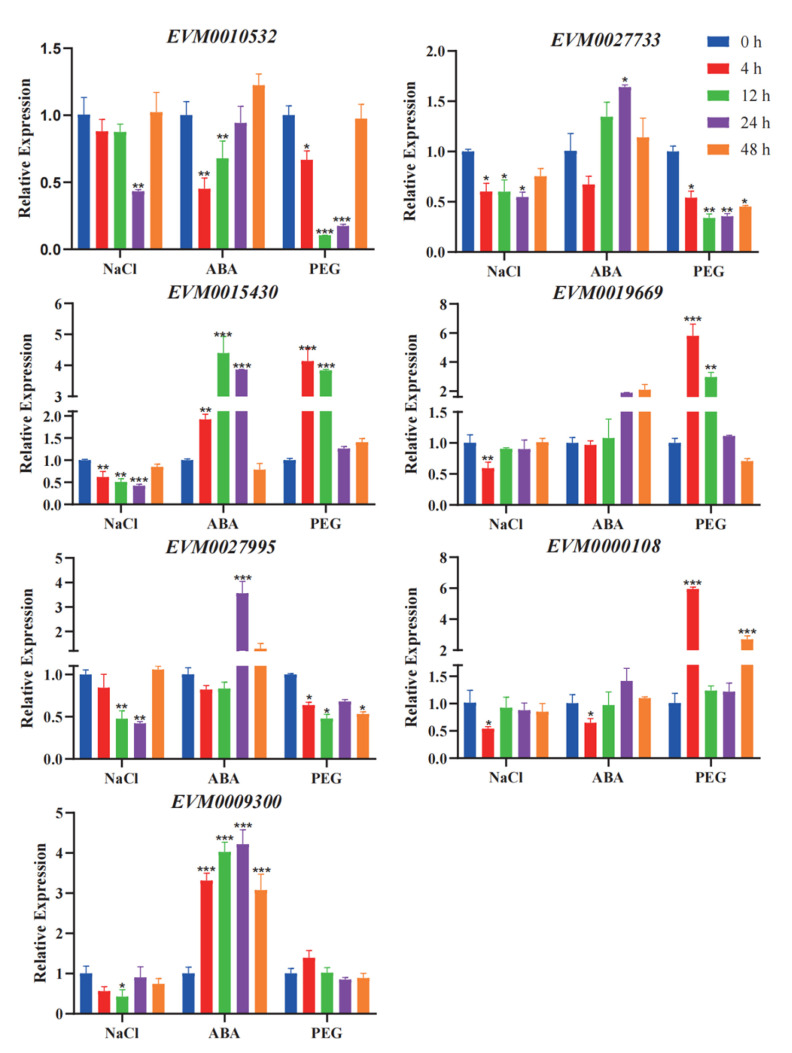
Relative expression levels of seven genes of WRKY, PHD, and MYB gene families verified by qRT-PCR under NaCl, ABA, and PEG stress treatments. Different colors in the histogram indicate stress treatment time at different time periods from 0 to 48 h. The error bars indicate the means ± SD (*n* = 3). Asterisks indicate a significant difference as determined by a Student’s *t*−test (* *p* < 0.05, ** *p* < 0.01, and *** *p* < 0.001).

**Table 1 genes-14-00463-t001:** Ka, Ks, and Ka/Ks statistics of VrWRKYs.

Gene ID	Gene ID	Ka	Ks	Ka/Ks	Gene ID	Gene ID	Ka	Ks	Ka/Ks
EVM0031994.1	EVM0021777.1	0.377257	1.355966	0.27822	EVM0019212.1	EVM0004727.1	0.494769	1.555822	0.318012
EVM0001250.1	EVM0010228.1	0.318741	0.71844	0.443657	EVM0016020.1	EVM0025273.1	0.210695	0.922716	0.228343
EVM0027287.1	EVM0008994.1	0.127341	1.112803	0.114433	EVM0007473.1	EVM0025273.1	0.323461	1.935039	0.16716
EVM0008820.1	EVM0019474.1	0.272679	0.805485	0.338528	EVM0027592.1	EVM0011639.1	0.288415	1.085786	0.265628
EVM0021285.1	EVM0020969.1	0.499739	2.310074	0.21633	EVM0024302.1	EVM0020494.1	0.2786	1.016181	0.274163
EVM0020620.1	EVM0017632.1	0.162551	0.814435	0.199587	EVM0014182.1	EVM0031025.1	0.495803	2.475382	0.200294
EVM0026717.1	EVM0002287.1	0.24299	0.971014	0.250244	EVM0009284.1	EVM0030301.1	0.13935	0.797171	0.174806
EVM0002826.1	EVM0003062.1	0.465142	5.29056	0.087919	EVM0029499.1	EVM0005845.1	0.174814	0.905242	0.193113
EVM0031593.1	EVM0026645.1	0.657167	1.357884	0.483964	EVM0027995.1	EVM0000159.1	0.515184	0.848937	0.606858
EVM0017859.1	EVM0001845.1	0.158668	0.775746	0.204536	EVM0020969.1	EVM0005039.1	0.337596	1.225598	0.275454
EVM0011623.1	EVM0008286.1	0.457181	2.158146	0.21184	EVM0003062.1	EVM0004390.1	0.276607	0.929997	0.297428
EVM0009282.1	EVM0008286.1	0.326785	0.903277	0.361778	EVM0001250.1	EVM0004390.1	0.504291	2.157071	0.233785
EVM0003197.1	EVM0029289.1	0.181362	0.521552	0.347734	EVM0008452.1	EVM0003477.1	0.297527	1.291118	0.230441
EVM0019669.1	EVM0017680.1	0.424596	1.498066	0.28343	EVM0004545.1	EVM0016786.1	0.362343	1.866473	0.194132
EVM0026625.1	EVM0002223.1	0.141808	0.598734	0.236846	EVM0014182.1	EVM0031269.1	0.23133	0.976648	0.236861
EVM0017131.1	EVM0024850.1	0.235397	0.771602	0.305076	EVM0031025.1	EVM0031269.1	0.499192	1.702519	0.293208
EVM0006656.1	EVM0013604.1	0.343103	2.769751	0.123875	EVM0011623.1	EVM0032969.1	0.257186	0.927936	0.277159
EVM0014182.1	EVM0004727.1	0.460905	4.070824	0.113222	EVM0022071.1	EVM0004772.1	0.176798	0.992875	0.178067
EVM0031025.1	EVM0004727.1	0.297638	0.753707	0.394898	EVM0021285.1	EVM0024302.1	0.524252	1.414758	0.37056
EVM0031269.1	EVM0004727.1	0.432249	1.959744	0.220564	EVM0014182.1	EVM0019212.1	0.442443	2.983577	0.148293

## Data Availability

Not applicable.

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
