# Peer review of "Identification and Functional Characterization of WRKY, PHD and MYB Three Salt Stress Responsive Gene Families in Mungbean (Vigna radiata L.)"

_genes, 2023, doi:10.3390/genes14020463_

Round 1

Reviewer 1 Report

Title: Title of manuscript is not appropriate so that need to be reframed.

Abstract: Abstract is not properly written in the current position; add some main results in the abstract.

Introduction:  Introduction parts are very simple and not have any mechanistic part in this manuscript.  Please mention your justification why need this type research. Reframe introduction part with recent references.  

Material and method:  Material and methods is proper.

Results: Results are not properly managed in the manuscript. Please try to improve whole manuscript results with the help of native English speaker.

Discussion: Discussion part is very simple, only description part discussed here in the currently. So that needs to improve/reframe whole manuscript and write discussion with mechanistic part along with recent references to support your results. 

Conclusion: Conclusion part is not appropriate and need to reframe with improvement. This part indicated only for your main results and future perspectives. So that essential need to improve whole manuscript with all section and including novelties in the abstract and conclusion part.    

Author Response

Respond to the first reviewer

Revision suggestions:

Question 1 Title: Title of manuscript is not appropriate so that need to be reframed.

Answer: Thanks very much for your comments and we have fixed the title “Genome-Wide Identification and Functional Characterization of WRKY, PHD and MYB three Family Members in Mungbean (Vigna radiata L.)” (lines 2-4).

Question 2 Abstract: Abstract is not properly written in the current position; add some main results in the abstract.

Answer: Thanks very much for your comments and we have fixed them and revised the related sentences in the revision (lines 12-31).

Question 3 Introduction: Introduction parts are very simple and not have any mechanistic part in this manuscript. Please mention your justification why need this type research. Reframe introduction part with recent references.

Answer: Thanks very much for your comments and we have fixed them and revised the related sentences in the revision (lines43-46, lines64, lines83-84, line117, lines135-136, lines152-154).

Question 4 Material and method:  Material and methods is proper.

Answer: Thanks very much for your comments. In the current version, we added the DNA methods in the revision (lines154-159).

Question 5 Material and method:  Material and methods is proper.

Answer: Thanks very much for your comments. In the current version, we added the DNA methods in the revision (lines159-161, line225).

Question 5 Results: Results are not properly managed in the manuscript. Please try to improve whole manuscript results with the help of native English speaker.

Answer: Thanks very much for your comments. In the current version, we had revised this part (lines154-159).

Question 6 Discussion: Discussion part is very simple, only description part discussed here in the currently. So that needs to improve/reframe whole manuscript and write discussion with mechanistic part along with recent references to support your results.

Answer: Thanks very much for your comments. In the current version, we had revised this part and deleted some paragraphs.

Question 7 Conclusion: Conclusion part is not appropriate and need to reframe with improvement. This part indicated only for your main results and future perspectives. So that essential need to improve whole manuscript with all section and including novelties in the abstract and conclusion part.

Answer: Thanks very much for your comments. In the current version, we had revised this part.

Reviewer 2 Report

References can be improved by removing the inappropriate reference from the text and the same in the list of reference. 

Author Response

Respond to the second reviewer

Additional comments.

Question 1 References can be improved by removing the inappropriate reference from the text and the same in the list of reference.

Answer: Thanks very much for your comments and we have fixed them and revised the related sentences in the revision.

Reviewer 3 Report

The manuscript is well-written and organized. It is within the journal's scope as it is an interesting and valuable topic. However, it contains numerous technical errors.

General remarks:

The article was not written in accordance with the journal's guidelines;

-Email addresses, as well as the author's initials, are not listed.

-The abstract is substantially longer than 200 words.

-The first keyword is bolded, which is incorrect.

-The citation of literature in the text must be corrected (superscript should not be used).

-In literature, journal citations should be in italics. 

- The font for tables and figures should be prepared by the journal's requirements.

-Duplication Type and Selection pressure should be removed from table 1, and they should be mentioned in the text in a single sentence. 

Author Response

Respond to the third reviewer

Comment:The manuscript is well-written and organized. It is within the journal's scope as it is an interesting and valuable topic. However, it contains numerous technical errors.

Revision suggestions:

Question 1 Email addresses, as well as the author's initials, are not listed.

Answer: Thanks very much for your comments and we have fixed them and revised the related sentences in the revision (lines7-11).

Question 2 The abstract is substantially longer than 200 words.

Answer: Thanks very much for your comments and we have fixed them and revised the related sentences in the revision (line13, 15, line24, line29, and line31).

Question 3 The first keyword is bolded, which is incorrect.

Answer: Thanks very much for your comments and we have fixed them and revised the related sentences in the revision (line32).

Question 4 The citation of literature in the text must be corrected (superscript should not be used).

Answer: Thanks very much for your comments and we have fixed them and revised the related sentences in the revision. 

Question 5 In literature, journal citations should be in italics.

Answer: Thanks very much for your comments and we have fixed them and revised the related sentences in the revision. 

Question 6 The font for tables and figures should be prepared by the journal's requirements.

Answer: Thanks very much for your comments and we have revised the related sentences in the revision.

Question 7 Duplication Type and Selection pressure should be removed from table 1, and they should be mentioned in the text in a single sentence.

Answer: Thanks very much for your comments and we have revised table 1 in the revision and lines 318. 

Reviewer 4 Report

Dear author; The manuscript is an important work that contains important information in general. I can suggest some revisions should be made.

Below are my revision suggestions:

- The purpose and justification of the study should be expressed more broadly.

- In the material and method section; The relevant document is not available at the web address where all protein sequences of mung bean are available. This can be a major issue and needs to be checked and fixed.

-Neighbor Joining is a method that does not have this problem. However, there is no attempt to determine what is the “best” tree. Usually, bootstrapping is performed to determine how frequently the various nodes of the dendrogram are represented, and these are indicated as numbers (out of x trees) or percent at the nodes of the dendrogram. Other means of determining confidence in dendrograms exist, and the authors need to provide a statistical basis for the choice of the most representative dendrogram.

Author Response

Respond to the fourth reviewer

Comment:The manuscript is an important work that contains important information in general. I can suggest some revisions should be made.

Revision suggestions:

Question 1 The purpose and justification of the study should be expressed more broadly.

Answer: Thanks very much for your comments and we have fixed them and revised the related sentences in the revision (lines 43-46; lines 83-84; lines 135-136, 152-154).

Question 2 In the material and method section; The relevant document is not available at the web address where all protein sequences of mung bean are available. This can be a major issue and needs to be checked and fixed.

Answer: Thanks very much for your comments and we have fixed them and revised the related sentences in the revision. All the data (Illumina, Nanopore, Hi-C and RNAseq) support the findings of this study are openly availablein the NCBI SRA (Sequence Read Archive) database under the Bioproject ID: PRJNA660308 (lines159-161).

Question 3 Neighbor Joining is a method that does not have this problem. However, there is no attempt to determine what is the “best” tree. Usually, bootstrapping is performed to determine how frequently the various nodes of the dendrogram are represented, and these are indicated as numbers (out of x trees) or percent at the nodes of the dendrogram. Other means of determining confidence in dendrograms exist, and the authors need to provide a statistical basis for the choice of the most representative dendrogram.

Answer: Thanks very much for your comments and we have fixed them and revised the related sentences in the revision.In this study, ClustalW program and MEGA7 software was used for multiple sequence alignment and evolutionary tree analysis. The calculation method was NJ (Poisson Correction), and the parameter was set to 1000 repeated iterations (the Bootstrap value of reliable evolutionary tree >70). We also tried to use MEGA7 in the Neighbor Joining analysis, the classification results were the same. As TBtools could did the collinearity analysis, we took the common analysis parameters in this study.

Round 2

Reviewer 4 Report

Dear author; Thank you for completing the revisions I have proposed. I suggest that the manuscript be published in this form in the journal.

Sincerely